# The Rise and Down of Babel Tower: Investigating the Evolution Process of Multilingual Code Large Language Model

Jiawei Chen[1,2], Wentao Chen[3], Jing Su[3], Jingjing Xu[3], Hongyu Lin[1,*] Mengjie Ren[1,2],
Yaojie Lu[1], Xianpei Han[1], Le Sun[1],
[1]Chinese Information Processing Laboratory, Institute of Software, Chinese Academy of Sciences
[2]University of Chinese Academy of Sciences
[3]ByteDance Inc.

## Abstract

Large language models (LLMs) have shown significant multilingual capabilities. However, the mechanisms underlying the development of these capabilities during pre-training are not well understood. In this paper, we use code LLMs as an experimental platform to explore the evolution of multilingual capabilities in LLMs during the pre-training process. Based on our observations, we propose the Babel Tower Hypothesis, which describes the entire process of LLMs acquiring new language capabilities. During the learning process, multiple languages initially share a single knowledge system dominated by the primary language and gradually develop language-specific knowledge systems. We then validate the above hypothesis by tracking the internal states of the LLMs through identifying working languages and language transferring neurons. Experimental results show that the internal state changes of the LLM are consistent with our Babel Tower Hypothesis. Building on these insights, we propose a novel method to construct an optimized pre-training corpus for multilingual code LLMs, which significantly outperforms LLMs trained on the original corpus. The proposed Babel Tower Hypothesis provides new insights into designing pre-training data distributions to achieve optimal multilingual capabilities in LLMs.

*A united human race speaking a single language migrates to Shinar where they agree to build a great city with a tower that would reach the sky. Yahweh, observing these efforts and remarking on humanity's power in unity, confounds their speech so that they can no longer understand each other and scatters them around the world, leaving the city unfinished.*

–The story of *Babel Tower*[1]

## 1 Introduction

Large language models (LLMs) have demonstrated remarkable multilingual capabilities (Philippy et al., 2023; Choenni et al., 2023), even in some low-resource languages (Lai et al., 2023; Chirkova & Nikoulina, 2024). Remarkably, LLMs are commonly pre-trained on multilingual corpora and demonstrate multilingual capabilities during the pre-training stage (Tanwar et al., 2023; Cahyawijaya et al., 2024). Consequently, an important question arises: how do the multilingual capabilities of LLMs evolve during pre-training process?

Many previous works have explored the multilingual mechanisms of LLMs. However, most of them are limited to investigating existing pre-trained LLMs rather than the dynamic process of

---

[*] Corresponding authors.
[1]https://en.wikipedia.org/wiki/Tower_of_Babel

pre-training. Some works have found that multilingual capabilities are similar to the translation process (Zhang et al., 2023). The LLMs form distinct language systems within their intermediate layers (Zhong et al., 2024; Wendler et al., 2024), and during generating, the LLMs transform the knowledge in these systems into target language tokens using language-specific neurons at top layers (Tang et al., 2024). Although these works explain the multilingual mechanism of existing LLMs, the formation of the mechanism during pre-training remains unexplored.

In this paper, we use code LLMs as an experimental platform to explore the evolution of multilingual capabilities in LLMs during the pre-training process (Li et al., 2023a; Guo et al., 2024). Specifically, we conducted preliminary experiments to analyze how monolingual LLMs learn a new language by continual pre-training the monolingual LLM on the corpus with the new language, similar to the setting of Zheng et al. (2024). As illustrated in Figure 1, our key finding is that the model's capabilities in newly added languages exhibit a pattern of rapid initial improvement, followed by a decline, and then stabilization (or slow improvement). Additionally, we observed that the point at which the capabilities of new languages begins to decline initially aligns closely with the point at which the knowledge transferred from the dominant language starts to decline. In the later stages of learning new languages, their performance gradually diverges from that of the dominant language.

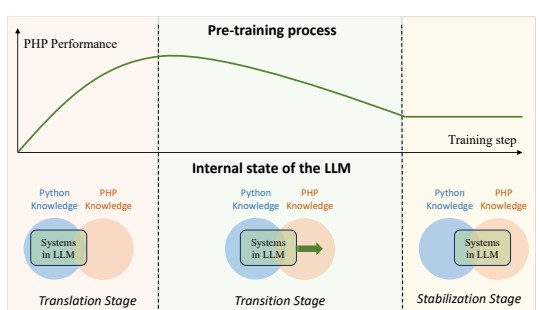

Figure 1: The evolution of a LLM learning a new language. In this figure, Python serves as the initial dominant language, with PHP as the new language. The process consists of three distinct stages: (1) Translation Stage: the performance of PHP improves rapidly and the Python system is dominated in the LLM, and PHP generation primarily relies on the Python system; (2) Transition Stage: the performance of PHP begins to decline while it gradually forms its own system; (3) Stabilization Stage: the performance of PHP stabilizes, and the generation of PHP depends on its own system.

This phenomenon suggests that in the early stages of learning, the capabilities of new language entirely rely on the capabilities of the dominant language existing in the LLM. As training progresses, the performance of the new languages gradually diverges from that of the dominant language. Based on these findings, we propose the Babel Tower Hypothesis, which posits that *during the learning process of large language models (LLMs), multiple languages initially share a single knowledge system dominated by a primary language and then gradually shift to developing multiple language-specific knowledge systems as training in new languages progresses.* With Babel Tower Hypothesis, the process of LLMs learning a new language can be divided into three stages: (1) Translation Stage: The LLM extensively relies on the dominant language knowledge system to response to the new language. (2) Transition Stage: The knowledge system of the new language is gradually established, and the generation of the new language increasingly transitions from the dominant language system to the new language system. (3) Stabilization Stage: The new language's responses primarily depend on its own system. To validate Babel Tower Hypothesis, we further track the internal states of the LLMs at various stages by identifying working languages (Wendler et al., 2024) and language transferring neurons (Tang et al., 2024) and found that the internal state changes of the LLM are consistent with our hypothesis: the generation of new language tokens progressively shifts from translating the primary language to utilizing its own system.

Additionally, our experiments have revealed that the establishment of new language knowledge systems does not necessarily lead to improved performance in those languages. In other words, for many languages, relying on a strong dominant language and translating its knowledge is more effective than building a new knowledge system using their own data. Building on these insights, we propose a novel method to construct the pre-training corpus for multilingual code LLM pre-training to achieve optimal performance. Specifically, we utilize the new language acquisition experiments to establish the relationship between performance and pre-training data distribution across different languages. By estimating the language distribution that corresponds to optimal performance, we can construct a pre-training corpus with an optimal distribution. Experimental results show that the code LLMs pre-trained with our optimized pre-training corpus significantly outperform those pre-trained with the original corpus by a large margin. Therefore, the proposed Babel Tower Hypothesis

provides a new insight into how we should design the data ratios for different languages during the pre-training stage to achieve optimal multilingual capabilities in LLMs.

Generally speaking, the contributions of this paper are:

- We propose the Babel Tower Hypothesis to explain the evolution process of multilingual LLM during pre-training: during the learning process of large language models (LLMs), multiple languages initially share a single knowledge system dominated by a primary language and then gradually shift to developing multiple language-specific knowledge systems as training in new languages progresses.

- We validate the Babel Tower Hypothesis in code LLMs by employing two methods to probe and analyze the internal states of LLMs during pre-training. Meanwhile, we revealed that for many languages, relying on a strong dominant language and translating its knowledge is more effective than building a new knowledge system using their own data.

- We design an effective method to construct a pre-training corpus with optimal distribution across different languages, providing a guide for the pre-training process.

## 2 RELATED WORK

Previous works mainly focused on studying the multilingual mechanisms within existing pre-trained LLMs. The primary approach they employ involves designing experiments to observe the generated outputs or internal states of these models. The primary methods of investigation can be categorized into three types: (1) analyzing the generation of the model: on the one hand, LLM can leverage knowledge acquired in various languages to generate outputs (Chen et al., 2022; Zhang et al., 2023; Kew et al., 2023; Yao et al., 2024). On the other hand, in many cases, the consistency of generated content across different languages remains low (Ohmer et al., 2023; Qi et al., 2023; Gao et al., 2024). (2) analyzing the output of intermediate layers: Wendler et al. (2024) use the logit lens method (Nostalgebraist, 2020) to detect the output tokens of intermediate layers in LLaMA-2 (Touvron et al., 2023). They discovered that these layers primarily operate in English, generating English tokens in the intermediate layers, even when the final output tokens are in other languages. Additionally, Zhong et al. (2024) found that the working languages are related to pre-training corpora. For instance, after continual pre-training using a Japanese corpus, LLaMA consistently produces Japanese tokens at the intermediate layers during token generation. (3) analyzing the activation states of neurons: some works identified language-specific neurons in LLMs that will be activated exclusively when processing specific languages (Tang et al., 2024; Kojima et al., 2024). Tang et al. (2024) found that these identified neurons are predominantly located in the top and bottom layers of the LLMs, serving to transform input into a unified semantic space or to transform the unified semantic space into output. Zhang et al. (2024) identify the core region in the parameters of LLMs that will influence the multilingual alignment and generalization, and they also observe that monolingual regions that will influence the ability of specific languages. In contrast to previous works, we investigate the evolution process of code LLMs during pre-training.

## 3 THE EVOLUTION PROCESS OF MULTILINGUAL SYSTEMS

In this section, we investigate the evolution of multilingual capabilities in LLMs during pre-training. Specifically, we introduce the Babel Tower Hypothesis to explain how these capabilities develop and then validate this hypothesis by tracking the internal system state of the LLM using two methods.

Table 1: The statistics of pre-training corpus.

|  | PHP | Go | C# | C++ | Python |
|---|---|---|---|---|---|
| #Instances | 23M | 11M | 37M | 35M | 16M |
| #Tokens | 23B | 11B | 16B | 28B | 11B |

### 3.1 EXPERIMENTAL SETUP

In this paper, the pre-training corpus is collected from GitHub, and its statistics are presented in Table 1. We conducted experiments utilizing the GPT-2 architecture (1.3 billion parameters). We consider Python as the dominant language due to its widespread use in various scenarios. The majority of experiments are conducted on a Python monolingual LLM, which will be pre-trained on corpus of other languages such as PHP, C#, Go, or C++. The training step is set to 50,000 and the

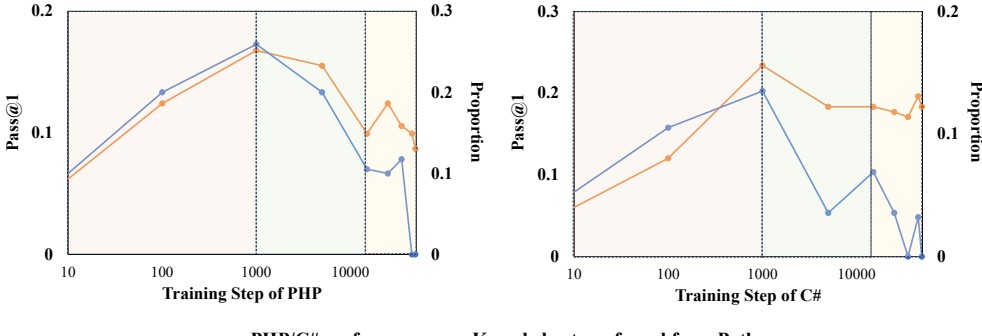

Figure 2: The performance of PHP/C# (left y-axis) and the proportion of correct answers requiring knowledge from the Python corpus (Knowledge transferred from Python, right y-axis) across training steps (x-axis). Based on the performance curve, we divide the entire process into three stages. The performance improves initially, then gradually declines, and eventually stabilizes. Concurrently, in the early stages, the proportion of correct answers requiring knowledge from the Python corpus is relatively high but subsequently decreases.

global batch size is 1,024. To trace the performance of Python, we incorporated a small amount of Python data (a total of 100 million tokens) into the monolingual corpus of other languages.

## 3.2 BABEL TOWER HYPOTHESIS

To investigate multilingual evolution during pre-training, we conducted a preliminary experiment to observe the process by which LLMs acquire new languages. Taking PHP and C# as an example, we analyze the performance variations in HumanEval (Chen et al., 2021) during pre-training. HumanEval is a problem-solving code evaluation benchmark focused on Python, which has been translated into various programming languages and is suitable for exploring the cross-language transfer abilities of LLMs (Zheng et al., 2023; Guo et al., 2024). To identify the sources of an LLM's capabilities, especially whether it leverages knowledge from the Python corpus during generation, we measure the proportion of correct answers requiring knowledge from the Python corpus. Specifically, we constructed a Python-specific subset of HumanEval that necessarily requires Python knowledge for solving by collecting problems that the Python monolingual LLM could correctly solve but which fail to be correctly solved by PHP or C# monolingual LLMs. Formally, let set $\mathbb{P}$ be the Python-specific subset and set $\mathbb{C}$ be the questions correctly solved by the LLM. The proportion of correct answers requiring knowledge from the Python corpus is calculated as $\frac{|\mathbb{P} \cap \mathbb{C}|}{|\mathbb{C}|}$.

The experimental result is shown in Figure 2. We can see that based on performance curve, the pre-training process can be divided into three stages:

(1) In the initial stage of pre-training, the performance of PHP and C# improved significantly. Concurrently, the proportion of correct answers necessarily relying on Python knowledge was notably high. We hypothesize that the LLMs are utilizing their Python knowledge to generate solutions for PHP and C# problems. The internal processing of the LLM may be analogous to translation; hence, we term this the **translation stage**.

(2) Subsequently, the performance gradually declined, accompanied by a decrease in the proportion of correct answers necessarily relying on Python knowledge. This could be because the PHP/C# system gradually formed, leading to a gradual transformation from the Python system to the PHP/C# system during generation. We term this the **transition stage**.

(3) Finally, the performance stabilized, there are almost no correct answers that necessarily require Python knowledge to solve. We hypothesize that the LLM might have reached a stable internal state dominated by PHP/C#. We term this the **stabilization stage**.

This phenomenon indicates that in the early stages of learning, the capabilities of a new language entirely rely on the capabilities of the dominant language existing in the LLM. Over time, as training progresses, the performance of the new languages gradually diverges from that of the dominant language. To this end, we propose the Babel Tower Hypothesis.

**Babel Tower Hypothesis**   *During the learning process of large language models (LLMs), multiple languages initially share a single knowledge system dominated by a primary language and then gradually shift to developing multiple language-specific knowledge systems as training in new languages progresses.*

The Babel Tower Hypothesis explains the evolution of multilingual LLMs during the pre-training process. We will further validate this hypothesis in the next sections.

### 3.3   INTERNAL STATE OF LLMS

To comprehensively understand the characteristics and differences among the three stages of the pre-training process in LLMs, we track the internal states of LLMs during pre-training using two distinct methods:

**(1) Working languages**   Similar to Wendler et al. (2024), we employ the logit lens (Nostalgebraist, 2020) to generate predicted tokens in intermediate layers, which can be seen as the working languages of the LLMs. In contrast to natural languages, the majority of programming languages are based on English, and a significant number of tokens overlap across different languages, making it difficult to identify the corresponding language simply through generated tokens. To accurately determine the working language, we use built-in functions or statements with identical functionalities but different tokens as identifiers. For example, to return the length of an array, Python uses the "*len()*" function, while PHP uses the "*count()*" function. When the "*count()*" function is generated in PHP, we can easily identify that an internally generated "*len*" is a Python token. In the end, we identified five such identifiers for each new language and utilized GPT-4o to generate ten code completion problems for each identifier, guiding the LLM to generate the corresponding tokens. When these identifiers are generated, we employ the logit lens method to identify the corresponding tokens generated in the intermediate layers (excluding the last five layers). Specifically, we count the number of generated tokens in each language and calculate the frequency of for each language. This frequency is used to estimate the proportion of working language. Formally, for language $i$, the proportion of it being the working language is calculated as follows:

$$\mathcal{R}_i = \frac{\epsilon_i}{\sum_j \epsilon_j} \tag{1}$$

where $\mathcal{R}_i$ the proportion of $i$ being the working language and $\epsilon_i$ is the number of generated tokens of language $i$ in the intermediate layers. The work language can measure the internal system of the model. When generating in the new language, if the Python system is dominant in the LLM, then the working language will primarily be Python.

**(2) Language transferring Neurons**   Based on Tang et al. (2024), there are language-specific neurons which will be activated in response to particular languages, and they are typically distributed across both the bottom and top layers in the LLM, representing the transformation process from input to intermediate layers and from intermediate layers to output. Therefore, we refer to these neurons as language transferring neurons. We will investigate the variations in the number of language transferring neurons during pre-training. Specifically, following Tang et al. (2024), we calculate the Language Activation Probability Entropy (LAPE) for each neuron, and identify language transferring neurons for each language. The number of language-transferring neurons per language serves as the final metric. When the primary language system dominates the LLM, a large number of neurons will be involved in language transfer for other languages. This is because these language transferring neurons are required to facilitate the transformation from intermediate layers to top layers and from bottom layers to intermediate layers.

The experimental results are shown in Figure 3 and 4, we summarize the characteristics of the internal states of LLMs during the pre-training process as follows:

**Translation Stage**   In the early stage of pre-training, Python serves as the primary working language, with a significantly higher number of language transferring neurons activated for the new language. This indicates that the LLM activates extra neurons to transform Python knowledge into tokens of the new language, akin to a translation process.

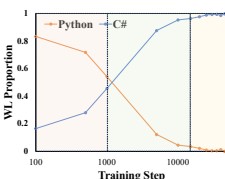 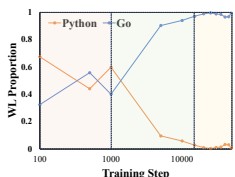 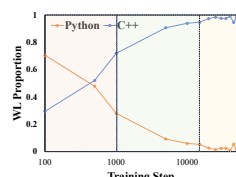

Figure 3: The proportion of a language being the working language (WL proportion, y-axis) when generating new languages across training steps (x-axis). Throughout the pre-training process, the working languages gradually transition from Python to the new languages.

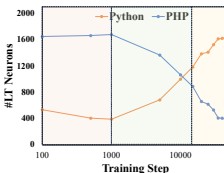 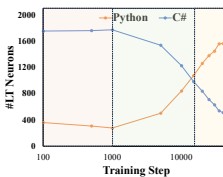 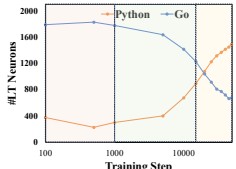 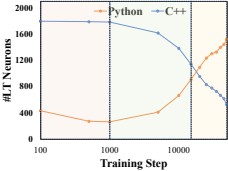

Figure 4: The number of language transferring neurons in different languages (#LT neurons, y-axis) across training steps (x-axis). Throughout the pre-training process, the language transferring neurons for new languages activated gradually diminish, whereas those for Python activated progressively increase.

**Transition Stage**  Throughout the pre-training process, the working language gradually transitions from Python to the new language, and the proportion of neurons related to new languages decreases accordingly. This indicates that the Python system gradually shifts to the new language system during the entire pre-training process.

**Stabilization Stage**  In the final stage, the LLMs mainly work on the new languages, with the number of language transferring neurons associated with the new language continues to decrease.

This demonstrates the process of system shifting during pre-training: from a single language system that shares knowledge across multiple languages to forming multiple systems for different languages.

## 4    ACHIEVING OPTIMAL MULTILINGUAL PERFORMANCE

Based on Figure 2, the performance peak occurs during the translation stage. This suggests that, for PHP and C#, leveraging the Python system is more effective than building their own systems. To achieve optimal performance, an intuitive approach would be to stop pre-training when the translation stage is reached. However, in practical multilingual pre-training, the pre-training corpus contains various languages that cannot reach their optimal stages simultaneously. As discussed in subsequent sections, C++ reaches optimal performance during the stabilization stage. Therefore, the early-stop strategy cannot be employed. To this end, we attempt to adjust the distribution of pre-training data in different languages to enable different languages to achieve optimal states concurrently. In this section, we analyze the impact of pre-training data distribution on performance and internal states of LLMs and propose a simple yet effective method to estimate the optimal pre-training data distribution.

### 4.1    THE IMPACT ON PRE-TRAINING DATA DISTRIBUTION

To understand the impact of pre-training data distribution on the performance and internal states of LLMs, we pre-train the Python monolingual LLM with a mixed corpus of Python and a new language. By varying the proportions of these languages within the pre-training corpus, we observed different final performances and internal states.

### 4.1.1    EXPERIMENTAL SETUP

In this section, we conduct experiments on PHP and will extend the conclusions to other languages in Section 4.2. The experimental setup is the same as in Section 3.1.

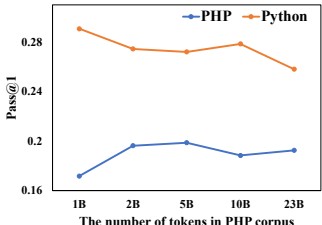 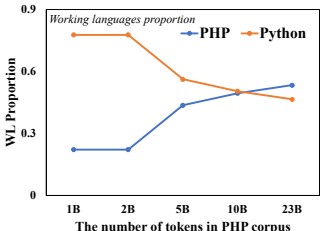 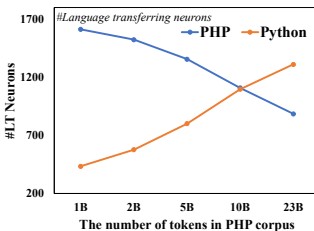

(a) The HumanEval Performance. (b) The proportion of a language being the working language (left, WL As the number of tokens in the PHP Proportion) and the number of language transferring neurons (right, corpus increases, the performance #LT Nerons). As the number of tokens in the PHP corpus increases, of PHP does not show correspond- the proportion of PHP as a working language also increases and the ing improvement. number of language transferring neurons for PHP declines.

Figure 5: The final internal states under different pre-training corpus. The X-axis represents the number of PHP tokens in the pre-training corpus, while the Y-axis indicates the performance, the proportion of the working languages, and the number of language transferring neurons corresponding to the final state.

**Pre-training Corpus** We downsample the original PHP pre-training corpus (23B tokens) and construct different PHP pre-training corpora based on the number of tokens, including 1B, 2B, 5B, and 10B. We then mix the PHP corpus with the Python corpus for pre-training.

**Metrics** We focus on the performance and internal state of the LLM. Specifically, we report three metrics using the last checkpoint of pre-training process: HumanEval performance on PHP, the proportion of PHP used as the working language, and the number of language transferring neurons for PHP. Given the variability in HumanEval performance, we obtain the final performance by calculating the average performance over a sliding window with a width of five, conducting evaluations every 1,000 steps.

### 4.1.2 EXPERIMENTAL RESULT

The experimental results are presented in Figure 5. We can see that:

**The pre-training data distribution across different languages determines the internal state of the LLM at the Stabilization Stage.** As shown in Figure 5b, when the number of PHP tokens in the pre-training corpus increases, the proportion of PHP in the working languages gradually rises, while the number of language transferring neurons for PHP decreases. This indicates that increasing the number of PHP tokens will ultimately lead to a higher proportion of PHP system in the LLMs.

**Achieving the optimal pre-training data distribution across different languages is crucial for effective cross-lingual transfer.** As shown in Figure 5a, increasing the number of PHP tokens initially enhances performance but subsequently leads to a decline. This trend indicates that, for HumanEval, the LLM benefits from the Python system for cross-language transfer. Therefore, achieving optimal performance requires establishing the optimal internal state within the LLM. Given the unpredictable natural distribution of different languages, utilizing all collected tokens for pre-training does not necessarily yield positive results.

The results indicate that the establishment of PHP knowledge system with more data does not necessarily enhance performance in it. Instead, reliance on the dominant Python system proves to be more effective. Compared to Figure 2, we observed a similar trend in PHP performance. This suggests that by adjusting the distribution of different languages in the corpus, it is possible to emulate the internal state observed at peak performance in Figure 2.

### 4.2 METHOD TO CONSTRUCT PRE-TRAINING CORPUS

The above experiments indicate that adjusting the pre-training data distribution in different languages can achieve an optimal performance at the stabilization stage. In this section, we propose a method for estimating the optimal pre-training data distribution.

---

**Algorithm 1** Estimate the corpus size of target language for optimal performance

---

**Input:** LLM $\mathcal{M}$; Checkpoint save interval $i$; The target language corpus $\mathbb{T}$; The validation set $\mathbb{V}$; The number of tokens in the Python corpus $\eta_{\text{python}}$;
**Output:** The number of tokens in the target language corpus;

---

1: Continual pre-training $\mathcal{M}$ with $\mathbb{T}$, saving checkpoints every $i$ steps and recording the corresponding training losses $\{\ell_1, \ell_2, \ldots, \ell_n\}$;
2: Recording the initial loss $\alpha$ and the lowest loss $\beta$;
3: Evaluating the saved checkpoints using $\mathbb{V}$ to obtain the performance set $\{s_1, s_2, \ldots, s_n\}$, and determine the index $j = \arg\max\{s_1, s_2, \ldots, s_n\}$;
4: Estimating the proportion of the Python system $\mathcal{P}(\ell_j)$ using $\ell_j, \alpha, \beta$ based on Equation 2;
5: Estimating the number of tokens in the target language corpus using $\mathcal{P}(\ell_j), \eta_{\text{python}}$ based on Equation 3, i.e., $\eta_{\text{target}} = \eta_{\text{python}}/\mathcal{P}(\ell_j) - \eta_{\text{python}}$;

---

To determine the relationship between overall performance and the pre-training data distribution across different languages, we use the internal states of LLMs as mediators since they are related to both of them. Specifically, we divide the method into two steps: (1) constructing the relationship between internal states and overall performance and (2) constructing the relationship between internal states and the pre-training data distribution.

### 4.2.1 RELATIONSHIP BETWEEN INTERNAL STATES AND OVERALL PERFORMANCE

Considering the similar trends in PHP performance depicted in Figures 2 and 5a, we establish a correlation between internal states and overall performance based on the new language acquisition experiment in Section 3.2. In the experiment, we transformed a Python monolingual LLM into monolingual LLM of a new language. We hypothesize that the proportion of the Python system decreases from 100% to 0%, while the proportion of the new language system increases from 0% to 100%. Consequently, we can use the observed performance changes during the pre-training process to estimate relationship between internal states and overall performance.

Specifically, we use training loss as a metric to measure system proportions for the following reasons: (1) it exhibits trends akin to those of working language metrics, characterized by continuous increases or decreases and an initially rapid rate of change that later decelerates; and (2) it demands less computational effort compared to working language metrics:

$$\mathcal{P}(\ell) \approx \frac{\ell - \beta}{\alpha - \beta} \tag{2}$$

where $\mathcal{P}(\ell)$ is the estimated proportion of the Python system at the step whose loss is $\ell$. $\alpha$ is the initial loss of LLM. Due to the excessively high value of the initial loss, the average loss of steps 90-100 is selected as the initial loss. $\beta$ is the final loss of LLM, and we use the lowest loss in pre-training as the final loss.

### 4.2.2 RELATIONSHIP BETWEEN INTERNAL STATES AND DISTRIBUTION

We use an intuitive approach to construct the relationship between internal states and the pre-training data distribution across different languages: using the proportion of a language tokens in the corpus as the final system proportion:

$$\bar{\mathcal{P}}(\eta_i) \approx \frac{\eta_i}{\sum_j \eta_j} \tag{3}$$

where $\bar{\mathcal{P}}(\eta_i)$ is the estimated proportion of the system of language $i$, and $\eta_i$ is the number of tokens of language $i$ in the pre-training corpus.

### 4.2.3 OVERALL PROCESS

To fully utilize the knowledge of Python for cross-language transfer, in this paper, we fix the Python corpus and control corpus size of others languages in corpus. Based on the above methods, the optimal number of tokens corresponding to a target language can be estimated. The specific process is shown in Algorithm 1.

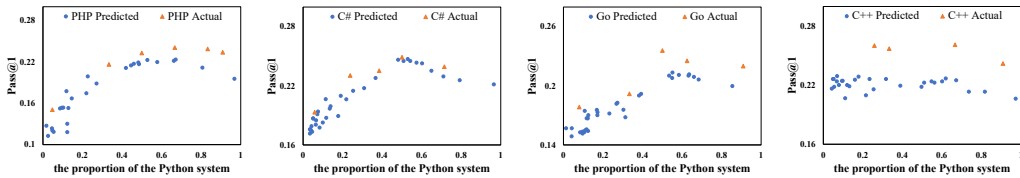

Figure 6: The average HumanEval performance of Python and the new language varies with the proportion of the Python system. Blue dots indicate the predicted results, while orange triangles represent the actual results obtained from pre-training.

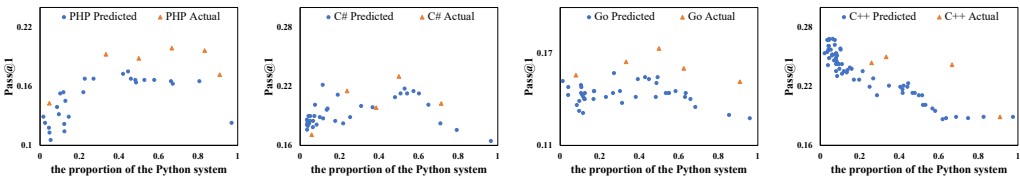

Figure 7: The HumanEval performance of the new language varies with the proportion of the Python system. Blue dots indicate the predicted results, while orange triangles represent the actual results obtained from pre-training.

In this way, we are able to determine the optimal number of tokens in the pre-training corpus of other languages, thereby achieving better overall performance. Next, we will conduct experiments to validate the proposed method.

### 4.3 EXPERIMENT

#### 4.3.1 SETUP

The experimental setup is the same as Section 3.1, and the new languages are PHP, C#, Go, or C++. We use the average HumanEval performance of Python and the new language and the performance of the new language as the final performance. In addition, we validate the estimated results by pre-training it on corpora with varying proportions. Notably, for C++, we observed that it relies less on the Python system. Consequently, we verified the results at only a few key points.

#### 4.3.2 EXPERIMENTAL RESULT

The experimental results are shown in Figure 6 and 7. Despite potential numerical discrepancies, the trend between the predicted and actual results remains consistent, emphasizing the effectiveness of our method. For most languages, as the proportion of Python integration increases, the predicted final performance initially rises and then declines. This pattern indicates that the new language can utilize the knowledge of the Python system for generation. However, when the proportion of the new language is excessively high, it may compete with the Python system, resulting in decreased performance. Conversely, a significantly low proportion of the new language in the model hinders its ability to generate the new language effectively. For C++, we found that it does not depend on the Python system and can achieve optimal performance through its own system. Our method is applicable to such languages, indicating its universality.

#### 4.3.3 PRE-TRAINING FROM SCRATCH

To further demonstrate the effectiveness of our method, we pre-train multilingual LLMs with 1.3 billion and 6.7 billion parameters from scratch using a mixture of corpora from five different languages. We compare the original corpora with our optimized versions. In addition to HumanEval (Chen et al., 2021), we evaluate the LLMs using the MBXP benchmark (Athiwaratkun et al., 2022). Specifically, based on Figure 6, we randomly down-sampled the token quantity for PHP and C# to 5 billion and 10 billion tokens, respectively, while keeping the token quantities for the other languages unchanged. The final metrics were derived from the average HumanEval and MBXP performances across all five languages. For 1.3B LLM, the total number of processed tokens is set to 500 billion. For 6.7B LLM, the total number of processed tokens is set to 800 billion.

Table 2: The HumanEval and MBXP performance (Pass@1) of different languages during pre-training. We compare the performance of LLMs pre-trained using the original corpus (Original) with those pre-trained using our optimized corpus (Optimized). The evaluated LLMs contain 1.3 billion (1.3B) and 6.7 billion (6.7B) parameters. AVE is the average performance of HumanEval and MBXP. Bold indicates the best results, and underline indicates the second-best results.

| | HumanEval | | | | | | MBXP | | | | | |
|---|---|---|---|---|---|---|---|---|---|---|---|---|
| | Python | PHP | C# | Go | C++ | AVE | Python | PHP | C# | Go | C++ | AVE |
| *1B+ Models* | | | | | | | | | | | | |
| StarCoder-1B (Li et al., 2023b) | 15.17 | 13.04 | 15.19 | 8.44 | 14.63 | 13.29 | 29.80 | 18.56 | 23.29 | 58.93 | 26.18 | 31.35 |
| StarCoder-3B (Li et al., 2023b) | 21.46 | 21.12 | 16.46 | 14.29 | 24.39 | 19.54 | 39.80 | 30.13 | 39.27 | 74.78 | 35.91 | 43.98 |
| DeepseekCoder-1.3B (Guo et al., 2024) | 29.88 | 23.60 | 27.22 | 20.13 | **31.10** | 26.39 | **52.40** | 45.41 | **48.40** | 78.35 | **55.11** | **55.93** |
| Original-1.3B (ours) | 28.66 | 24.22 | 23.42 | **21.43** | 27.44 | 25.03 | 41.20 | 33.41 | 43.38 | **79.24** | 45.39 | 49.81 |
| Optimized-1.3B (ours) | **32.93** | **26.08** | **29.75** | 16.88 | 27.44 | **26.61** | 45.40 | 46.29 | 40.64 | 77.90 | 49.63 | 51.97 |
| *6B+ Models* | | | | | | | | | | | | |
| StarCoder-7B (Li et al., 2023b) | 28.37 | 24.22 | 27.85 | 17.53 | 25.61 | 24.72 | 46.20 | 42.36 | 45.66 | 71.88 | 43.64 | 49.95 |
| StarCoder2-7B (Lozhkov et al., 2024) | 35.40 | 32.30 | 39.24 | 22.08 | 34.76 | 32.76 | 53.13 | 55.02 | 54.79 | 82.37 | 55.61 | 60.18 |
| CodeLlama-7B (Rozière et al., 2024) | 33.50 | 24.22 | 34.81 | 20.13 | 27.44 | 28.02 | 49.80 | 43.89 | 48.40 | 72.32 | 48.63 | 52.61 |
| DeepseekCoder-6.7B (Guo et al., 2024) | 46.95 | 37.89 | **47.47** | **31.82** | **44.51** | 41.73 | **65.80** | **66.16** | **66.67** | **89.78** | **67.58** | **71.20** |
| Original-6.7B (ours) | 46.95 | 38.51 | 39.24 | 26.62 | 40.24 | 38.31 | 59.00 | 57.86 | 61.19 | 83.04 | 60.60 | 64.34 |
| Optimized-6.7B (ours) | **48.17** | **41.61** | 44.94 | 31.17 | 43.90 | **41.96** | 63.60 | 59.83 | 61.42 | 81.03 | 64.84 | 66.14 |

The experimental results are shown in Table 2. Compared to the original pre-training corpus, the corpus optimized with our method exhibits a significant average performance improvement, indicating that our approach is suitable for a broader range of pre-training scenarios. Furthermore, our method remains effective even when applied to an LLM with 6.7 billion parameters, indicating the universality of our approach. In future work, our method could be integrated with the filtering of pre-training corpora to construct higher-quality pre-training data.

## 5 DISCUSSIONS

**Analogy to second language acquisition**   The development of multilingual capabilities in large language models (LLMs) resembles second language acquisition (SLA) in humans (Krashen, 1981). During the learning process, systems for various languages gradually take shape (Cook, 2016). Additionally, the acquisition of a second language is influenced by previously mastered languages, resulting in language transfer (Lightbown & Spada, 2021). This correlation between multilingual learning in LLMs and human language acquisition may enable the application of human cognitive theories to explain various behaviours of LLMs in the future.

**The significance of this paper**   Firstly, this paper reveals the mechanisms underlying the multilingual capabilities of LLMs during the pre-training process. It provides guidance for multilingual pre-training, such as data proportioning. Additionally, our findings indicate that pre-training strategies may differ across languages. For example, for some low-resource languages, thinking mainly within the primary language system rather than constructing an entirely new system might be the optimal strategy. Lastly, the findings highlight potential limitations of previous multilingual LLMs (Cui et al., 2023; Csaki et al., 2024; Faysse et al., 2024; Zheng et al., 2024). For instance, using a specific language's corpus for continual pre-training does not necessarily enhance the LLM's performance.

## 6 CONCLUSIONS

In this paper, to explain the evolution process of multilingual capabilities in LLMs during pre-training, we propose **Baber Tower Hypothesis**: *during the learning process of large language models (LLMs), multiple languages initially share a single knowledge system dominated by a primary language and then gradually shift to developing multiple language-specific knowledge systems as training in new languages progresses*. Taking code LLMs as an example, we employ two methods to validate the hypothesis. We further found that for some languages, leveraging a dominant language and transferring knowledge from it is often more effective than building a new knowledge system with additional pre-training data. Finally, we propose a simple and effective method to estimate the optimal distribution of pre-training corpora across different languages. In the future, our approach can be extended to natural language scenarios and scaled to larger models.

## 7 ACKNOWLEDGE

We sincerely thank the reviewers for their insightful comments and valuable suggestions. This work was supported by Beijing Natural Science Foundation (L243006), the Basic Research Program of ISCAS (Grant No. ISCAS-JCZD-202303, ISCAS-ZD-202402).

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

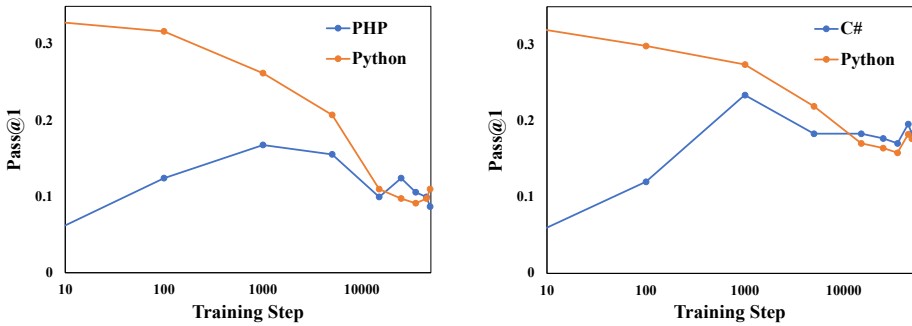

Figure 8: The performances of Python and PHP/C# (y-axis) across training steps (x-axis).

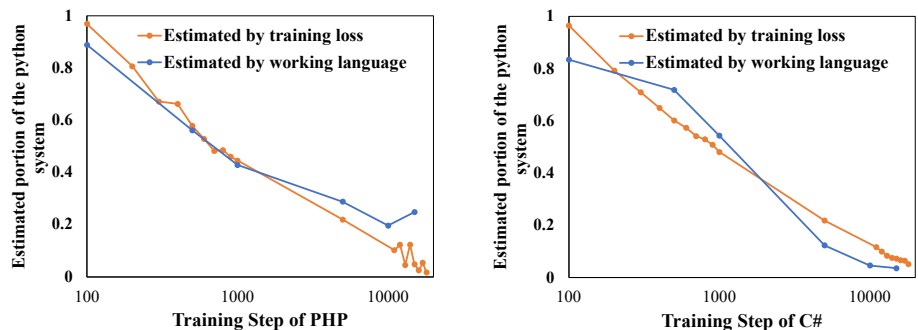

Figure 9: The estimated proportion of Python system (y-axis) across training steps (x-axis). The orange line represents the proportion of the Python system estimated using the training loss, i.e., equation 2. The blue line represents the proportion of the Python system using the proportion of Python as the working language.

# A   APPENDIX

## A.1   WORKING LANGUAGES DETECTION

To determine the language of the generated tokens, we use built-in functions or statements with identical functionalities but different tokens as identifiers. The identifiers are shown in Figure 3. When the PHP identifier "strlen" is generated, we will use logit lens method to generate tokens in the intermediate layers. The generated "len" will be considered as Python while "strlen" will be considered as PHP.

Table 3: The identifiers for detecting the working languages and the corresponding Python tokens.

| PHP | "strlen": "len"; "count": "len"; "sort": "sorted"; "gettype": "type"; "array_sum": "sum"; |
|---|---|
| C# | "Console.WriteLine": "print"; "Console.ReadLine": "input"; ".IndexOf": ".find",".index"; "catch": "except"; ".Add": ".append"; |
| Go | "=append": ".append"; "true": "True"; "false": "False"; "func": "def"; "&&": "and"; "\|\|": "or"; "!": "not"; "nil": "None"; |
| C++ | ".push_back": ".append"; "catch": "except"; "&&": "and"; "\|\|": "or"; "!": "not"; ".erase": ".pop"; "to_string": "str"; |

## A.2   THE PERFORMANCE OF PYTHON

We report the Python performance in the experiments in Section 3.2. As shown in Figure 8, in the first stage, the performance of the new language increases, while Python experiences a slight decline. During the second stage, as the new language system is established and competes with Python, Python's performance significantly declines. Finally, Python's performance stabilizes.

## A.3 THE ESTIMATED PROPORTION OF THE PYTHON SYSTEM

To further demonstrate the relationship between the training loss and estimated proportion of the python system, we used working language for verification since the working language can reflect the internal state of the LLM. Specifically, we calculate the proportion of Python as the working language at each checkpoint and assume that this can be approximated to the proportion of Python systems. We then compare this result with the proportion of Python systems estimated from the corresponding training loss. The results are presented in Figure 9. We can see that the proportion of the Python system estimated from the training loss closely approximates that estimated from the working language, thereby substantiating the relationship between training loss and the estimated proportion of the Python system.

