# OpenReview forum: "The Rise and Down of Babel Tower: Investigating the Evolution Process of Multilingual Code Large Language Model"
_ICLR.cc/2025/Conference — ICLR 2025 Poster_

### Official Review · Reviewer_4Qay · 2024-11-02

**Soundness:** 3
**Presentation:** 2
**Contribution:** 3
**Rating:** 6
**Confidence:** 4

**Summary:**

This work focuses on LLM multilingual capability (in terms of programming language instead of natural language). The authors observed a 3-stage division during the continual learning setting (i.e. pre-train on Python, and continue pre-training on PHP/C#...): (1) translation stage, where the performance of the new language improves and the Python system is dominated in the LLM and the generation in the new language primarily relies on the Python system; (2) Transition Stage: the performance of the new language begins to decline while it gradually forms its own system; (3) Stabilization Stage: the performance of the new language stabilizes, and the generation in the new language depends on its own system. Based on such observation, the authors proposed the “Babel Tower Hypothesis” which depicts the language learning process in multilingual pre-training - a single knowledge system was first formed, followed by forming language-specific systems.

To support this claim, the authors used 2 methods to evaluate the internal state of the LLMs during pre-training, (1) working languages and (2) language transferring neurons. These metrics show similar 3-stage trends. Finally, the authors proposed to adjust the pre-training data distribution to make different languages to achieve optimal states concurrently. Empirical results show that this optimized data distribution outperforms the original data distribution in terms of the HumanEval and MBXP benchmarks.

**Strengths:**

1. The authors investigated the training dynamics for different data setup, which is intriguing.
2. I like the idea of checking LLM internal states using "working languages" and "Language transferring Neurons" concepts.
3. Experiments are solid.

**Weaknesses:**

1. The presentation of Algorithm 1 and subsequent experiments and results in Section 4.3.1 and 4.3.2 needs some clarification. The last line in Algorithm 1 states that the number of tokens in the target language is computed using $P(l_j)$ and $\eta_{python}$ based on Eqn 3. However, Eqn. 3 does not have a $P(l_j)$ term.
2. Assuming the estimated num of tokens are obtained, I am still confused about the results in Fig. 6 and Fig. 7. For example, what do “PHP predicted” and “PHP Actual” mean? My understanding is that you are estimating the data proportion in pre-training corpora, not the performance on HumanEval. Why there are multiple results for multiple data proportions (i assume there should be an optimal one given by Algorithm 1.) And why there are much more “PHP predicted” dots than “PHP Actual”? I may have misunderstand something, but i think the presentation of these sections could be made clearer.

**Questions:**

1. Section 4.3.3 states that the optimized pre-training data distribution is formed by “we randomly down-sampled the token quantity for PHP and C# to 5 billion and 10 billion tokens, respectively, while keeping the token quantities for the other languages unchanged ”. I’m quite surprised that training on a smaller amount of data provides such a big performance improvement in Table 2. What is the optimized data distribution?  The models are trained on the data collected by the authors themselves (as described in Section 3.1). Have you performed any checks on data quality? Is it possible that these results are actually due to some bad quality data (e.g. noisy/incorrect codebases), instead of the claimed knowledge transfer between languages?
2. I think a better set of experiments may be to use existing open-source multilingual datasets, and show that the proposed optimized data distribution can produce stronger pre-train models. In this case the comparison can include existing coding LLMs.
3. Further, programming languages are very different from natural languages due to their highly structured form. Based purely on programming languages, it is not clear whether the “Babel Tower Hypothesis” is generalizable or not. It may be more informative if there are more experimental results on natural languages to support this general hypothesis - though, I understand the difficulty due to resource demands for these experiments - a smaller scale of multilingual training data is understandable.

---

> ### Author Response · Authors · 2024-11-22
> **Authors' response (1/2)**
>
> Thanks for your constructive comment. Here is our response to your concerns.
> > [W1] How to compute the number of tokens in the target language using $P(\ell_j)$ and $\eta_{\text{python}}$ based on Eqn 3 ?
>
> In Algorithm 1, after computing $P(\ell_j)$ using Equation 2, we assign its value to $\bar{P}(\eta_j)$ in Equation 3, since both $P(\ell_j)$ and $\bar{P}(\eta_j)$  represent the estimated proportion of the Python system. The number of tokens corresponding to the new language can be calculated based on Equation 3:
>  $$\eta_{\text{target}}=\eta_{\text{python}}/\mathcal{P}(\ell_j)-\eta_{\text{python}}$$
> Thank you for your suggestion. We provide a clearer description in the revision.
>
> > [W2]  What do “PHP predicted” and “PHP Actual” mean in Figure 6 and 7? And why there are much more “PHP predicted” dots than “PHP Actual”?
>
> Each point in Figure 6 and 7 represents one checkpoint, the y-axis is performance and the x-axis is the proportion of the Python system.
>
> "PHP predicted" contains one pre-training process as Algorithm 1, and the pre-training corpus is PHP corpus. We evaluate the checkpoints during the pre-training process and calculate the proportion of the Python system based on the corresponding training loss using Equation 2.
>
> "PHP actual" contains several pre-training processes as in Section 4.1, and the pre-training corpus is a mixed corpus of Python and PHP (with varying proportions for different pre-training processes). We evaluate the final checkpoint of each experiment and calculate the proportion of the Python system based on the corpus proportions of different languages using Equation 3.
>
> In one training process, “PHP predicted” contains many checkpoints  while "PHP actual" contains only one checkpoint, so the number of dots in "PHP predicted" is significantly higher than the number of points in "PHP actual".
>
> > [Q1] What is the optimized data distribution? Have you performed any checks on data quality? Is it possible that these results are actually due to some bad quality data (e.g. noisy/incorrect codebases), instead of the claimed knowledge transfer between languages?
>
> The optimized data distribution is estimated using Algorithm 1. In this paper, we utilize high-quality pre-training data, having conducted quality filtering and decontamination beforehand. We apply Algorithm 1 to estimate the number of tokens in the target language corpus for an optimized data distribution. As noted in Section 4.3.3, during down-sampling, we employ a purely random strategy without excluding poor-quality data.

---

> ### Author Response · Authors · 2024-11-22
> **Authors' response (2/2)**
>
> > [Q2] I think a better set of experiments may be to use existing open-source multilingual datasets, and show that the proposed optimized data distribution can produce stronger pre-train models. In this case the comparison can include existing coding LLMs.
>
> Thank you for your suggestion. Given the potential data quality issues inherent to open-source pre-training corpus, we use our own higher-quality data to mitigate the bias you mentioned above, i.e., performance variations arising from the inclusion or exclusion of low-quality data.
>
> We have pre-trained our models for an extended number of steps. For the 1.3B LLM, the total number of processed tokens was set to 500 billion, while for the 6.7B LLM, the total number of processed tokens was set to 800 billion. We compared our models (both Original and Optimized) with other open-source models, and the results are presented below:
>
> 1.3B HumanEval performance:
> |                       | Python | PHP   | C\#   | Go    | C++   | AVE   |
> |-----------------------|--------|-------|-------|-------|-------|-------|
> | StarCoder-1B          | 15.17  | 13.04 | 15.19 | 8.44  | 14.63 | 13.29 |
> | StarCoder-3B          | 21.46  | 21.12 | 16.46 | 14.29 | 24.39 | 19.54 |
> | DeepseekCoder-1.3B    | 29.88  | 23.6  | 27.22 | 20.13 | **31.1**  | 26.39 |
> | Original-1.3B (ours)  | 28.66  | 24.22 | 23.42 | **21.43** | 27.44 | 25.03 |
> | Optimized-1.3B (ours) | **32.93**  | **26.08** | **29.75** | 16.88 | 27.44 | **26.61** |
>
> 6.7B HumanEval performance:
>
> |                       | Python | PHP   | C\#   | Go    | C++   | AVE   |
> |-----------------------|--------|-------|-------|-------|-------|-------|
> | StarCoder-7B          | 28.37  | 24.22 | 27.85 | 17.53 | 25.61 | 24.72 |
> | StarCoder2-7B         | 35.4   | 32.3  | 39.24 | 22.08 | 34.76 | 32.76 |
> | CodeLlama-7B          | 33.5   | 24.22 | 34.81 | 20.13 | 27.44 | 28.02 |
> | DeepseekCoder-6.7B    | 46.95  | 37.89 | **47.47** | **31.82** | **44.51** | 41.73 |
> | Original-6.7B (ours)  | 46.95  | 38.51 | 39.24 | 26.62 | 40.24 | 38.31 |
> | Optimized-6.7B (ours) | **48.17**  | **41.61** | 44.94 | 31.17 | 43.90  | **41.96** |
>
> As the number of training steps increased, our model achieved performance that surpassed several open-source LLMs. Furthermore, the LLM pre-trained with optimized pre-training corpora (Optimized) outperformed the LLM pre-trained with original pre-training corpora (Original), demonstrating the effectiveness of our approach.
>
> > [Q3] Based purely on programming languages, it is not clear whether the “Babel Tower Hypothesis” is generalizable or not. It may be more informative if there are more experimental results on natural languages to support this general hypothesis.
>
> Code LLMs have gained significant attention from researchers and represent an important direction in current research. In this paper, although we focus on programming languages, we do not make additional assumptions about language distribution, thus enabling potential extensions to other distributions, such as natural language. Our validation methods, such as working language and language transfer neurons, are derived from related works in LLMs of natural language. Due to resource limitations, we plan to further explore the adaptation of our work in the context of natural language processing in the future.

---

> > ### Author Response · Authors · 2024-11-27
> > **Looking forward to your feedback**
> >
> > Dear reviewer,
> >
> > Thanks for your constructive comments! We have provided responses to your concerns. Do our responses address your questions? We are happy to discuss more if you still have any concerns. Thank you again and look forward to your feedback!

---

> ### Author Response · Authors · 2024-12-02
> **Looking forward to your feedback**
>
> Dear reviewer,
>
> The rebuttal discussion period is coming to a close and have provided responses to your concerns. We are happy to discuss more if you still have any concerns.

---

> ### Author Response · Authors · 2024-12-02
> **Looking forward to your feedback**
>
> Dear Reviewer 4Qay
>
> We hope this message finds you well. We apologize for reaching out repeatedly, but with less than two days remaining in the discussion period, we wanted to check whether our updates have addressed your concerns kindly.
>
> If you have any further questions or require additional clarification, we are more than happy to provide it. Your feedback has been invaluable, and we greatly appreciate your time and effort in reviewing our work.
>
> Thank you again for your continued support. We look forward to your feedback.
>
> Sincerely
>
> All authors

---

> > ### Comment · Reviewer_4Qay · 2024-12-02
> > **Thanks for rebuttal**
> >
> > I have raised my score since I think this paper could be an interesting contribution. Authors' rebuttal answered some of my questions.

---

### Official Review · Reviewer_cRyw · 2024-11-04

**Soundness:** 1
**Presentation:** 2
**Contribution:** 2
**Rating:** 3
**Confidence:** 4

**Summary:**

This submission explores how Large Language Models (LLMs) develop multilingual capabilities. The authors propose a three-stage learning process: translation, transition, and stabilization. They experimented with code LLMs by tracking the working languages and language transferring neurons during pre-training.

**Strengths:**

The mechanisms behind the development of multilingual capabilities during pre-training is an important area.

**Weaknesses:**

* The submission's use of the term "multilingual capabilities" is unclear. It seems to refer to the model's capacity to understand multiple programming languages, rather than the more common interpretation of "multilingual" as understanding multiple natural languages. Since the programming languages studied are all English-based, the method doesn't address the orthographic differences of various natural languages. Instead, the experiments reduced to focus on how the semantics of programming languages are developed during pre-training.

* The methodology's flaws undermine the hypothesis presented. The claim that multilingual programming language capabilities are primarily developed during pre-training is unconvincing. It's likely that fine-tuning plays a significant role, and a more comprehensive mechanism should be considered.

* Additionally, the assertion that Python is the dominant language due to its widespread use lacks evidence and requires further clarification.

**Questions:**

Why combine the concepts of multiple natural languages and multiple programming languages?

Note: Authors should not reveal their identities. Descriptions like "Specifically, we conducted preliminary experiments to analyze how monolingual LLMs learn a new language by continual pre-training the monolingual LLM on the corpus with the new language (Zheng et al., 2024)" should be revised to remove identifying information.

---

> ### Author Response · Authors · 2024-11-22
> **Authors' response**
>
> Thanks for your comment. Here is our response to your concerns.
>
> > [W1 and Q1] The submission's use of the term "multilingual capabilities" is unclear. It seems to refer to the model's capacity to understand multiple programming languages, rather than the more common interpretation of "multilingual" as understanding multiple natural languages. Why combine the concepts of multiple natural languages and multiple programming languages?
>
> In this paper, we explore the multilingual capabilities of code LLMs, analogous to their natural language counterparts, where "multilingual" in this context refers to multiple programming languages.
>
> "Multilingual" is a common expression used in code LLM and many existing open-source code LLMs use this term, such as CodeGeeX [1], StarCoder [2], and DeepseekCoder [3].
>
> [1] CodeGeeX https://github.com/THUDM/CodeGeeX
>
> [2] StarCoder https://huggingface.co/blog/starcoder
>
> [3] DeepseekCoder https://deepseekcoder.github.io/
>
> > [W2] The methodology's flaws undermine the hypothesis presented. The claim that multilingual programming language capabilities are primarily developed during pre-training is unconvincing. It's likely that fine-tuning plays a significant role, and a more comprehensive mechanism should be considered.
>
> Pre-training and fine-tuning are both essential stages for a LLM to acquire multilingual capabilities. This paper primarily focuses on the pre-training stage, as it is a critical stage in developing the multilingual capabilities of LLMs [1]. Understanding how LLMs achieve multilingual capabilities through unsupervised training constitutes an important research direction. And there are many related studies studying the multilingual abilities of pre-trained base LLMs [2, 3].
>
> [1] Language-Specific Neurons: The Key to Multilingual Capabilities in Large Language Models https://arxiv.org/abs/2402.16438
>
> [2] LLMs Are Few-Shot In-Context Low-Resource Language Learners https://arxiv.org/abs/2403.16512
>
> [3] Do Llamas Work in English? On the Latent Language of Multilingual Transformers https://arxiv.org/abs/2402.10588
>
> > [W3] The assertion that Python is the dominant language due to its widespread use lacks evidence and requires further clarification.
>
> The selection of the primary programming language is influenced by the inherent characteristics of the language. We chose Python due to its widespread popularity and the availability of a large corpus of high-quality code.
> Several related studies highlight Python's dominance in the field of code LLMs. For example, CodeLlama has released a python-specialized version, CodeLlama-70B-Python [1]. Additionally, existing evaluations of open-source models [2,3] and benchmarks for assessing open-source code predominantly emphasize Python [4,5].
>
> [1] https://ai.meta.com/blog/code-llama-large-language-model-coding/
>
> [2] StarCoder: may the source be with you!, https://arxiv.org/abs/2305.06161
>
> [3] Qwen2.5-Coder Technical Report https://arxiv.org/abs/2409.12186
>
> [4] BigCodeBench: Benchmarking Code Generation with Diverse Function Calls and Complex Instructions https://arxiv.org/abs/2406.15877
>
> [5] LiveCodeBench: Holistic and Contamination Free Evaluation of Large Language Models for Code https://arxiv.org/abs/2403.07974
>
> > Note: Authors should not reveal their identities. Descriptions like "Specifically, we conducted preliminary experiments to analyze how monolingual LLMs learn a new language by continual pre-training the monolingual LLM on the corpus with the new language (Zheng et al., 2024)" should be revised to remove identifying information.
>
> This sentence may potentially lead to some misunderstandings; we cite Zheng et al., (2024) to indicate that the setting of our preliminary experiments (in Section 3) is similar to the setting of the experiments in Zheng et al., (2024), i.e., learning a new language through continual pre-training. We will revise this sentence to avoid any misunderstanding.

---

> ### Author Response · Authors · 2024-11-27
> **Looking forward to your feedback**
>
> Dear reviewer,
>
> Thanks for your constructive comments! We have provided responses to your concerns. Do our responses address your questions? We are happy to discuss more if you still have any concerns. Thank you again and look forward to your feedback!

---

> ### Author Response · Authors · 2024-12-02
> **Looking forward to your feedback**
>
> Dear reviewer,
>
> The rebuttal discussion period is coming to a close and have provided responses to your concerns. We are happy to discuss more if you still have any concerns.

---

> ### Author Response · Authors · 2024-12-02
> **Looking forward to your feedback**
>
> Dear Reviewer cRyw
>
> We apologize for reaching out repeatedly, but with less than two days remaining in the discussion period, we wanted to check whether our updates have addressed your concerns kindly.
>
> If you have any further questions or require additional clarification, we are more than happy to discuss more. Your feedback has been invaluable, and we greatly appreciate your time and effort in reviewing our work.
>
> Thank you again and look forward to your feedback!
>
> Sincerely
>
> All authors

---

> > ### Author Response · Authors · 2024-12-02
> > **Looking forward to your feedback**
> >
> > Dear Reviewer cRyw
> >
> > We apologize for reaching out repeatedly, but with less than one day remaining in the discussion period,  we wanted to check whether our updates have addressed your concerns kindly.
> > We greatly appreciate your time and effort in reviewing our work, and look forward to your feedback!
> >
> > Sincerely
> >
> > All authors

---

### Official Review · Reviewer_rkZp · 2024-11-04

**Soundness:** 3
**Presentation:** 4
**Contribution:** 3
**Rating:** 6
**Confidence:** 3

**Summary:**

This paper investigates the multilingual capabilities of large language models (LLMs), specifically focusing on how a monolingual code LLM trained in Python can integrate additional programming languages, such as PHP, C#, Go, and C++.

Analyzing the model's performance on code problem-solving tasks, the study proposes a hypothesis, termed the "Babel Tower Hypothesis," suggesting that learning new languages involves three phases:
 1. translation
 2. transition
 3. stabilization

The hypothesis is tested through an analysis of working languages and language-transferring neurons.
Additionally, the paper proposes a method to construct an optimal distribution of languages during pre-training to enhance multilingual performance.

**Strengths:**

1. This paper is clearly written, making it easy to read and follow.

2. The study’s perspective and findings are intriguing, with well-designed experiments and thorough analysis that convincingly support the conclusions.

3. It achieves good code-LLM perfermance by experimenting with pre-training data distribution.

**Weaknesses:**

1. The scope of this paper is a bit limited, as it focuses exclusively on code LLMs. Due to the unique characteristics of programming languages, the findings may not be easily applicable to natural languages.

2. The scenario explored is somewhat simplistic, focusing mainly on extending a primary LLM (Python in this case) to additional languages. It remains unclear whether the hypothesis would still hold if a different primary language were used.

3. Some parts of this paper are unclear to me:
 - In sections 4.2.1 and 4.2.2, it is not well explained how $P(l)$ relates to $\alpha$ and $\beta$ and how $P(l)$ determines the optimal number of tokens for the target languages.
 - Additionally, according to equation (2), the proportion $P(l)$ cannot exceed 1. What happens if there is sufficent amout of training data for the target language?
 - The hypothesis primarily addresses the extension of a dominant language to one additional language. However, in section 4.3.3, the strategy of training with five languages from scratch also appears to yield better performance. Why is this the case?

**Questions:**

Typo:
 - line 385 "Rrecoding" -> "Recording"

---

> ### Author Response · Authors · 2024-11-22
> **Authors' response (1/2)**
>
> Thanks for your suggestions on our work. Here is our response to your concerns.
>
> > [W1] The scope of this paper is a bit limited, as it focuses exclusively on code LLMs. Due to the unique characteristics of programming languages, the findings may not be easily applicable to natural languages.
>
> Code LLMs have gained significant attention and represent an important direction in current research [1,2,3]. Meanwhile, code LLM also supports numerous important applications, such as Cursor[4] and Copilot[5]. Therefore, we believe that code LLM itself is an important topic. In this paper, although we focus on programming languages, we do not make additional assumptions about language distribution, thus enabling potential extensions to other distributions, such as natural language. We plan to further explore the adaptation of our work in the context of natural language processing in the future.
>
> [1] StarCoder: may the source be with you!, https://arxiv.org/abs/2305.06161
>
> [2] Code Llama: Open Foundation Models for Code, https://arxiv.org/abs/2308.12950
>
> [3] Qwen2.5-Coder Technical Report， https://arxiv.org/abs/2409.12186
>
> [4] https://www.cursor.com/
>
> [5] https://github.com/features/copilot
>
> > [W2] It remains unclear whether the hypothesis would still hold if a different primary language were used.
>
> The selection of the primary programming language is influenced by the inherent characteristics of the language. We chose Python due to its widespread popularity and the availability of a large corpus of high-quality code.
>
> Several related studies highlight Python's dominance in the field of code LLMs. For example, CodeLlama has released a python-specialized version, CodeLlama-70B-Python [1]. Additionally, existing evaluations of open-source models [2,3] and benchmarks for assessing open-source code predominantly emphasize Python [4,5].
>
> [1] https://ai.meta.com/blog/code-llama-large-language-model-coding/
>
> [2] StarCoder: may the source be with you!, https://arxiv.org/abs/2305.06161
>
> [3] Qwen2.5-Coder Technical Report https://arxiv.org/abs/2409.12186
>
> [4] BigCodeBench: Benchmarking Code Generation with Diverse Function Calls and Complex Instructions
> https://arxiv.org/abs/2406.15877
>
> [5] LiveCodeBench: Holistic and Contamination Free Evaluation of Large Language Models for Code https://arxiv.org/abs/2403.07974
>
> > [W3-1] In sections 4.2.1 and 4.2.2, it is not well explained how the estimated proportion of the Python system P(l) relates to α ( the initial loss) and β (the final loss) and how P(l) determines the optimal number of tokens for the target languages.
>
> In Equation 2, we establish a linear relationship between the training loss $\ell$ and the proportion of the Python system $P(\ell)$, based on the initial loss $\alpha$ and final loss $\beta$. During pre-training, as the loss gradually decreases from \alpha to \beta, the corresponding proportion of the Python system is also gradually decreasing.
>
> To determine the optimal proportion, we apply Algorithm 1. Specifically, we pre-train the Python monolingual LLM with the corpus of a new language and save several checkpoints. Then we evaluate the saved checkpoints and identify the checkpoint with the best performance. Futhermore, we use the training loss of the best checkpoint to estimate the current proportion of Python system $P(\ell_j)$ (via Equation 2). After computing $P(\ell_j)$ using Equation 2, we assign its value to $P(\eta_j)$ in Equation 3, since both $P(\ell_j)$ and $P(\eta_j)$  represent the estimated proportion of the Python system. The number of tokens corresponding to the new language can be calculated based on Equation 3: $\eta_{\text{target}}=\eta_{\text{python}}/\mathcal{P}(\ell_j)-\eta_{\text{python}}$.
>
> > [W3-2] According to equation (2), the proportion P(l) cannot exceed 1. What happens if there is sufficient amount of training data for the target language?
>
> With the addition of more training data, the corresponding $\beta$ (the final training loss during pre-training) will change, ensuring that it does not exceed 1.

---

> > ### Author Response · Authors · 2024-11-22
> > **Authors' response (2/2)**
> >
> > > [W3-3] The hypothesis primarily addresses the extension of a dominant language to one additional language. However, in section 4.3.3, the strategy of training with five languages from scratch also appears to yield better performance. Why is this the case?
> >
> > Training with five languages can be seen as training with one majority language and multiple minority languages. The relationship between a majority language and a single minority language is essentially the same as that between a majority language and multiple minority languages.
> >
> > There are potential influences among various minority languages. However, our experimental results indicate that such influence is minimal. We conducted three sets of experiments:
> >
> > A. Training from scratch using a single minority language
> >
> > B. Training from scratch using Python in combination with a single minority language
> >
> > C. Training from scratch using Python in combination with multiple minority languages, as described in Section 4.3.3.
> >
> > For the first two sets of experiments, the total number of processed tokens for pre-training is set to 100 billion. For the third experiment, due to the inclusion of a broader corpus, the total number of processed tokens is set to 200 billion.
> > We evaluated the performance using the HumanEval dataset. To mitigate the impact of random fluctuations, we report the average performance of the final three checkpoints.
> >
> > |     | Exp A: PHP/C# | Exp B: Python+PHP/C# | Exp C: Python+PHP+C#+Go+C++ |
> > |-----|---------------|----------------------|-----------------------------|
> > | PHP | 10.15         | 16.56                | 16.15                       |
> > | C#  | 13.92         | 20.01                | 22.78                       |
> > | AVE | 12.04         | 18.29 (+6.25)        | 19.47 (+1.18)               |
> >
> > The text in parentheses indicates the performance changes of Experiment B relative to Experiment A, and of Experiment C relative to Experiment B.
> >
> > The results indicate a significant difference between Experiment A and Experiment B, indicating that Python contributes substantially to the performance of minority languages. However, the difference between Experiment B and Experiment C is relatively small, indicating that the impact among different minority languages is less significant compared to the influence of Python.
> >
> > > About typo
> >
> > Thank you for your suggestion. We will refine it in the revision.

---

> > > ### Comment · Reviewer_rkZp · 2024-11-30
> > >
> > > Thank the author for the clarification. After considering the explanation and the feedback from other reviewers, I lean toward accepting this paper, as I find the findings particularly interesting. Therefore, I will maintain my current score.

---

> ### Author Response · Authors · 2024-11-27
> **Looking forward to your feedback**
>
> Dear reviewer,
>
> Thanks for your constructive comments! We have provided responses to your concerns. Do our responses address your questions? We are happy to discuss more if you still have any concerns. Thank you again and look forward to your feedback!

---

### Official Review · Reviewer_6AVA · 2024-11-06

**Soundness:** 3
**Presentation:** 2
**Contribution:** 3
**Rating:** 6
**Confidence:** 3

**Summary:**

This work proposes a Babel tower hypothesis for the evolution of multi-lingual code LLM, that is continual pre-training a monolingual code LLM to master a new language will go through translation, transition, and stabilization stages. The authors demonstrate the evolution process of a python LLM based on GPT2 for new languages PHP and C#, revealing the best performance achieved at the translation stage.  Based on the observation the authors propose an algorithm to estimate the optimal distribution of the training corpus for different languages based on a linear hypothesis of the training loss and the monolingual system. The authors conduct experiments for new languages PHP, C#, Go, and C++ for HumanEval and MBXP, suggesting better performance compared to a baseline data distribution.

**Strengths:**

1. Overall this paper is well organized and the topic is interesting. The authors justify their hypothesis in a clear way.

2. The authors make a strong analogy of the learning process to human acquisition of new languages, and present some well-designed results including the evolution process and the transition of the internal states of LLM. Also, they apply the transition for optimization of corpus distribution, which could be significant for training of multi-lingual LLMs.

**Weaknesses:**

1. The paper lacks some detailed demonstration of the crucial parts of their claims. Please refer to my questions.

2. When presenting the methodology, I believe the authors make some important assumptions and analogy without further justification, which might make the presentation less persuasive. Please also refer to my questions for details.

**Questions:**

1. In Fig 2 the authors show the evolution of performance of new languages, however, it is also important to show the performance of Python to illustrate the impact on the learned languages.

2. Could the authors show some example of python-specific problems which are crucial to define the knowledge transferring procedure?

3. The authors define a linear relationship between the training loss and estimated portion of the python system as in eq (2). Could the authors clarify: (1) how to calculate the loss which might consist of python data only; and (2) further justification of the relationship other than results in Fig 6 and 7 to demonstrate the scalability of this hypothesis.

4. If I understand it correctly, the authors assume there is no intervening impact between different language during learning procedure. Both the evidences to demonstrate the learning stages of the new language and the proposed optimal data distribution are base on this assumption. Could the authors show some solid justification for this assumption?

5.  The authors only present the results with a baseline distribution in Table 2. I wonder if it is comparable to other optimized data distributions.

---

> ### Author Response · Authors · 2024-11-22
> **Authors' response (1/3)**
>
> Thanks for your constructive comments. We appreciate your feedback and here is our response to your concerns.
>
> > [Q1] It is also important to show the performance of Python to illustrate the impact on the learned languages.
>
> As illustrated in Section 3.1,  to trace the performance of Python, we incorporated a small amount of Python data (a total of 100 million tokens) into the monolingual corpus of other languages. The result is shown in the table and we also included a figure (Figure 8)  in the revision of our paper:
>
> | Training step | PHP         | Python       |
> |---------------|-------------|--------------|
> | 100           | 0.124223602 | 0.317073171  |
> | 1000          | 0.167701863 | 0.262195122  |
> | 5000          | 0.155279503 | 0.207317073  |
> | 15000         | 0.099378882 | 0.109756098  |
> | 25000         | 0.124223602 | 0.097560976  |
> | 35000         | 0.105590062 | 0.091463415  |
> | 45000         | 0.099378882 | 0.097560976  |
>
> In the first stage, the performance of the new language increases, while Python experiences a slight decline. During the second stage, as the new language system is established and competes with Python, the Python's performance significantly declines. Finally, Python's performance stabilizes.
>
> > [Q2] Could the authors show some examples of python-specific problems which are crucial to define the knowledge transferring procedure?
>
> Python-specific problems primarily encompass tasks that Python excels at, often depending on the extent of related knowledge within the pre-training corpus. For example, compared to PHP, Python usually has more algorithm corpus and thus demonstrates superior performance in algorithmic challenges, such as "parentheses matching" and "computing polynomial derivatives".
>
> > [Q3-1] How to calculate the loss which might consist of python data only in Equation 2.
>
> As illustrated in Algorithm 1, Equation 2 is applied in the new language acquisition experiment. In this experiment, we pre-train a Python monolingual LLM with the corpus of the new language. Consequently, the training loss is computed on the new language data rather than Python.
>
> > [Q3-2] The authors define a linear relationship between the training loss and estimated portion of the python system as in eq (2).
>
> Further justification of the relationship other than results in Fig 6 and 7 to demonstrate the scalability of this hypothesis
> To further demonstrate the relationship between the training loss and the proportion of the python system, we conducted a working language analysis to trace the internal state of the LLM.
>
> As illustrated in Section 3.3, working language is the language which is generated in the intermediate layers of LLMs. We can use the proportion of Python being the working language as an approximation for the proportion of the Python system.
>
> Specifically, we calculate the proportion of Python being the working language at different training steps during the pre-training process. Then we compare them with the proportion of Python systems estimated from the training loss (Equation 2). The result is shown in Figure 9 of the revision of our paper and Figure 9 shows that the relationship between the training step and the Python system exhibits a logarithmic-linear correlation.
>
> We used observation data (take PHP as an exmaple) to fit the relationship between training step and the proportion of the Python system:
>
> Training loss: $y_1=-0.3758*log(x)+1.6166$
>
> Working language: $y_2 = -0.3328*log(x) + 1.4973$
>
> where x represents the training step, $y_1$ represents the proportion of the Python system estimated using training loss, and $y_2$ represents the proportion of the Python system estimated using the working language.
>
> We calculate the difference between the two: $\delta = y_1-y_2 = -0.043 * log(x) + 0.1193$, where $\delta$ represents the difference between the proportion of the Python system we use to estimate the training loss and the corresponding proportion for which Python is used as the working language.
>
> | training step | $\delta$        |
> |---------------|---------------|
> | 100           | 0.0333        |
> | 200           | 0.02035571    |
> | 300           | 0.012783786   |
> | 400           | 0.00741142    |
> | 500           | 0.00324429    |
> | 1000          | -0.0097       |
> | 2000          | -0.02264429   |
> | 3000          | -0.030216214  |
> | 4000          | -0.03558858   |
> | 5000          | -0.03975571   |
> | 10000         | -0.0527       |
>
> We can see that the difference between estimates using training loss and those using working language is minimal, thereby substantiating the relationship between training loss and the estimated proportion of the Python system.

---

> > ### Author Response · Authors · 2024-11-22
> > **Authors' response (2/3)**
> >
> > > [Q4] the authors assume there is no intervening impact between different language during learning procedure. Both the evidence to demonstrate the learning stages of the new language and the proposed optimal data distribution are based on this assumption. Could the authors show some solid justification for this assumption?
> >
> > All languages influence one another theoretically, but our paper does not assume that there are no influences between other languages. However, there are plenty of experimental and experiential evidences suggesting that majority languages have the most significant impact on minority languages. For instance, many previous works focused on the relationship between one majority language and other minority languages [1,2,3]. At the same time, our experimental results demonstrate that the influences between minority languages are minimal. Here are our ablation study experiments:
> >
> > A. Training from scratch using a single minority language (PHP or C#)
> >
> > B. Training from scratch using Python in combination with a single minority language
> >
> > C. Training from scratch using Python in combination with multiple minority languages, as described in Section 4.3.3.
> >
> > For the first two sets of experiments, the total number of processed tokens for pre-training is set to 100 billion. For the third experiment, due to the inclusion of a broader corpus, the total number of processed tokens is set to 200 billion.
> > We evaluated the performance using the HumanEval dataset. To mitigate the impact of random fluctuations, we report the average performance of the final three checkpoints.
> >
> > |     | Exp A: PHP/C# | Exp B: Python+PHP/C# | Exp C: Python+PHP+C#+Go+C++ |
> > |-----|---------------|----------------------|-----------------------------|
> > | PHP | 10.15         | 16.56                | 16.15                       |
> > | C#  | 13.92         | 20.01                | 22.78                       |
> > | AVE | 12.04         | 18.29 (+6.25)        | 19.47 (+1.18)               |
> >
> > The text in parentheses indicates the performance changes of Experiment B relative to Experiment A, and of Experiment C relative to Experiment B.
> >
> > The results indicate a significant difference between Experiment A and Experiment B, indicating that Python contributes substantially to the performance of minority languages. However, the difference between Experiment B and Experiment C is relatively small, indicating that the impact among different minority languages is less significant compared to the influence of Python.
> >
> > This experiment further demonstrates that the influences between other minority languages are minimal.
> >
> > [1] Breaking Language Barriers: Cross-Lingual Continual Pre-Training at Scale https://arxiv.org/abs/2407.02118
> >
> > [2] Multilingual Pretraining and Instruction Tuning Improve Cross-Lingual Knowledge Alignment, But Only Shallowly  https://arxiv.org/abs/2404.04659
> >
> > [3] Don't Trust ChatGPT when Your Question is not in English: A Study of Multilingual Abilities and Types of LLMs https://arxiv.org/abs/2305.16339

---

> > > ### Author Response · Authors · 2024-11-22
> > > **Authors' response (3/3)**
> > >
> > > > [Q5] The authors only present the results with a baseline distribution in Table 2. I wonder if it is comparable to other optimized data distributions.
> > >
> > > Due to the high computational cost associated with the pre-training stage, searching the optimal data proportion through exhaustive search is challenging. Currently, data proportions for open-source code LLMs are typically determined based on original natural distributions [1,2] (as our baseline). This paper introduces a straightforward yet effective method to identify the optimal data proportion, marking one of its core contributions.  We further compare our method with open-source LLMs. Despite differences in experimental settings, our method demonstrates strong results, thereby proving its effectiveness.
> > >
> > > We have pre-trained our models for an extended number of steps. For the 1.3B LLM, the total number of processed tokens was set to 500 billion, while for the 6.7B LLM, the total number of processed tokens was set to 800 billion. We compared our models (both Original and Optimized) with other open-source code LLMs, and the results are presented below:
> > >
> > > 1.3B HumanEval performance:
> > >
> > > |                       | Python | PHP   | C\#   | Go    | C++   | AVE   |
> > > |-----------------------|--------|-------|-------|-------|-------|-------|
> > > | StarCoder-1B          | 15.17  | 13.04 | 15.19 | 8.44  | 14.63 | 13.29 |
> > > | StarCoder-3B          | 21.46  | 21.12 | 16.46 | 14.29 | 24.39 | 19.54 |
> > > | DeepseekCoder-1.3B    | 29.88  | 23.6  | 27.22 | 20.13 | **31.1**  | 26.39 |
> > > | Original-1.3B (ours)  | 28.66  | 24.22 | 23.42 | **21.43** | 27.44 | 25.03 |
> > > | Optimized-1.3B (ours) | **32.93**  | **26.08** | **29.75** | 16.88 | 27.44 | **26.61** |
> > >
> > > 6.7B HumanEval performance:
> > >
> > > |                       | Python | PHP   | C\#   | Go    | C++   | AVE   |
> > > |-----------------------|--------|-------|-------|-------|-------|-------|
> > > | StarCoder-7B          | 28.37  | 24.22 | 27.85 | 17.53 | 25.61 | 24.72 |
> > > | StarCoder2-7B         | 35.4   | 32.3  | 39.24 | 22.08 | 34.76 | 32.76 |
> > > | CodeLlama-7B          | 33.5   | 24.22 | 34.81 | 20.13 | 27.44 | 28.02 |
> > > | DeepseekCoder-6.7B    | 46.95  | 37.89 | **47.47** | **31.82** | **44.51** | 41.73 |
> > > | Original-6.7B (ours)  | 46.95  | 38.51 | 39.24 | 26.62 | 40.24 | 38.31 |
> > > | Optimized-6.7B (ours) | **48.17**  | **41.61** | 44.94 | 31.17 | 43.90  | **41.96** |
> > >
> > > As the number of training steps increased, our model achieved performance that surpassed several open-source LLMs. Furthermore, the LLM pre-trained with optimized pre-training corpora (Optimized) outperformed the LLM pre-trained with original pre-training corpora (Original), demonstrating the effectiveness of our approach.
> > >
> > > [1] DeepSeek-Coder: When the Large Language Model Meets Programming - The Rise of Code Intelligence (https://arxiv.org/pdf/2401.14196)
> > >
> > > [2] StarCoder: may the source be with you! (https://arxiv.org/pdf/2305.06161)

---

> ### Author Response · Authors · 2024-11-27
> **Looking forward to your feedback**
>
> Dear reviewer,
>
> Thanks for your constructive comments! We have provided responses to your concerns. Do our responses address your questions? We are happy to discuss more if you still have any concerns. Thank you again and look forward to your feedback!

---

> ### Author Response · Authors · 2024-12-02
> **Looking forward to your feedback**
>
> Dear reviewer,
>
> The rebuttal discussion period is coming to a close and have provided responses to your concerns. We are happy to discuss more if you still have any concerns.

---

> > ### Comment · Reviewer_6AVA · 2024-12-02
> > **Thanks for the rebuttal**
> >
> > The authors addressed some of my concerns and thus I raised my score. Good luck.

---

### Author Response · Authors · 2024-11-22
**Common responses**

We thank all reviewers for the insightful feedback. We have made the following revisions to our paper:

+ Revised the sentences from lines 61 to 67 to avoid misunderstanding on authors' identity disclosure. (Reviwer cRyw)

+ Revised typo: line 385 "Rrecoding" -> "Recording" (Reviewer rkZp)

+ Added an equation at line 390 (within Algorithm 1) that specifically illustrates the calculation of the number of tokens in the target
language corpus. (Reviwer 4Qay W1)

+ Included Python performance results in Appendix 2, shown in Figure 8. (Reviewer 6AVA Q1)

+ Added experiments in Appendix 3 to further demonstrate the relationship between training loss and the proportion of the Python system, shown in Figure 9. (Reviwer 6AVA Q3)

---

### Meta-Review · Area_Chair_Hf6t · 2024-12-08

**Metareview:**

This work focuses on the multilingual capabilities of LLMs (in the context of programming languages rather than natural languages). The authors observed a three-stage progression during continual learning (i.e., pre-training on Python, followed by continued pre-training on other programming languages): translation, transition, and stabilization. Based on these observations, the authors proposed the "Babel Tower Hypothesis," which describes the language learning process in multilingual pre-training. According to this hypothesis, a unified knowledge system is initially formed, which is later followed by the development of language-specific systems. Finally, the authors proposed adjusting the pre-training data distribution to enable different languages to achieve optimal states simultaneously. Empirical results show that this optimized data distribution outperforms the original distribution. Most of the concerns were addressed during the rebuttal.

**Additional Comments On Reviewer Discussion:**

Reviewer rkZp pointed out the limitation of focusing only on programming languages. While other aspects are worth investigating, this alone is not a valid reason to reject the paper.

Reviewer cRyw highlighted the issue of the use of the term ''multilingual'', but based on authors' rebuttal, it is a widely accepted term in the context of code LLMs.

Reviewer cRyw and Reviewer rkZp questioned the decision to use Python as the primary language in the experiments. Although exploring other languages as primary options would be valuable, this is not a valid reason for rejecting the paper.

The remaining comments pertain to clarification questions.

---

### Decision · Program_Chairs · 2025-01-22

Accept (Poster)